# Air Filter and Selected Vehicle Characteristics

**František Synák \*, Alica Kalašová and Ján Synák**

Department of Road and Urban Transport, University of Žilina, 01026 Žilina, Slovakia;
alica.kalasova@fpedas.uniza.sk (A.K.); synak5@stud.uniza.sk (J.S.)
**\*** Correspondence: frantisek.synak@fpedas.uniza.sk; Tel.: +42-1907-230-469

**Abstract:** While a vehicle is driving, the air filter is gradually being clogged. Thus, there is a condition for air mass delivered into the engine's cylinders to be reduced. It can further degrade the vehicle's ability to accelerate and, therefore, have an impact on road driving safety, or it can eventually make a vehicle performance worse or cause a change in the exhaust gas composition. The article pays attention to the impact of clogged air filter on selected vehicle characteristics including a vehicle's dynamic features, fuel consumption and composition of the exhaust gases. The measurements have been performed under laboratory conditions. For higher result objectivity, the measurements have been done under some extreme conditions as well, with both the air filter fully clogged and forced air induction. The importance of the article lies in quantification of the impact of clogged air filter on selected vehicle features. The results from this article show a relatively low impact of air filter on the features studied. The biggest differences were measured under extreme conditions when the air filter had been almost fully clogged, or the air had been delivered into the engine under pressure.

**Keywords:** air filter; air pollution; combustion engine; emissions; exhaust gases; fuel consumption; torque power

## 1. Introduction

The European Union is committed to achieving carbon neutrality by 2050 [1]. In order to achieve this, the transport sector must make some contributions as well, since it is responsible for around a third of all $CO_2$ emissions [2]. Combustion engine vehicles using the fossil fuels are most represented in road driving [3]. The amount of fuel consumed has an effect on the amount of $CO_2$ produced. Each litre of petrol consumed causes a production of at least 2.5 kg of $CO_2$, while diesel oil is 2.7 kg of $CO_2$ [4]. Besides $CO_2$, combustion engine vehicles are also producers of many other harmful emissions affecting the environment and human health [5]. The composition of exhaust gases varies depending on conditions of fuel consuming as well [6].

In order for the combustion engine to be active, a constant air input is needed. The intake air contains particles and other impurities that could cause early engine wear as well as fouling the sensors in the engine intake manifold [7,8]. That is why the air needed for burning is being transported into the engine cylinders through the air filter. Its role is to catch the unwanted impurities as much as possible and, at the same time, have minimum losses during the air flowing. These particles from the intake air are caught on the air filter that is, therefore, gradually being clogged [9]. According to [10], the air filter is, in many cases, replaced by the new one too late, leaving it fouled significantly. On the other hand, there are many air filters replaced too early with minimum foulness [11,12]. A clogged filter gives a precondition for relatively smaller air mass to be delivered into the cylinder [13,14]. Such an impact of the air filter on air mass delivered into the engine cylinders is being studied by several publications [15,16]. Many of these types of studies are realized via mathematical modelling, as can be seen in the publications [17–19]. This article pays attention to the impact of air filter on selected operational vehicle features.

However, besides the fact of how much is the air filter clogged, there are other factors that affect the amount of intake air. These are the engine design, method of air intakes, control unit programming and so on [20]. The impact of a clogged filter on selected vehicle features also depends on whether there is a compression ignition engine (CI), or a spark ignition engine (SI).

The power of CI engine is regulated qualitatively by composition of a mixture. That means the power is regulated by the amount of fuel injected into the air of nearly constant volume since the CI engine does not regulate the amount of air intake by a throttle valve. This regulation can be done by changing the turbocharger's management, if present, controlling the opening of the exhaust gas recirculation valve (EGR), timing the valves, etc. Concerning the compression ignition engine, attention paid to the richness of mixture is predominantly due to its impact on composition of the exhaust gas emissions. Thus, in many cases, when reducing the air mass delivered, there is also a reduced amount of fuel as a result of restriction of the engine's excessive smoke emission [21].

Concerning the spark ignition engine, the power is regulated qualitatively. To certain air mass corresponds a certain amount of fuel depending on the engine's working mode [22]. If the air filter is clogged, it can lead to reduction of intake air mass into the cylinder, and thus, to reduction of the amount of fuel injected [23]. Reduction of the amount of fuel injected leads to reduction of the amount of chemical energy from the fuel, the amount of converted chemical energy into thermal energy, and thus, to reduction of the amount of mechanical energy on the combustion engine's flywheel [24].

Most frequently, the engine control unit receives the information about the amount of intake air from the mass air flow sensor (MAF sensor), or the intake manifold absolute pressure sensor (MAP sensor) [25]. The engine control unit receives the information about the composition of a mixture by mass from the lambda sensor that is placed in the exhaust manifold [26].

Relating to both the SI engine and CI engine, it is presumed that when the air filter is clogged, the engine power and torque are reduced and the composition of exhaust gases together with specific fuel consumption are changed [27]. However, on the other hand, there are also tools that can eliminate the impact of clogged air filter on operational vehicle features. These include, for instance, changes in turbocharger's management, angle of throttle opening, timing of valve opening, etc. [28].

The purpose of measurements in this article is to determine the impact of a clogged air filter on selected vehicle characteristics. The measurements were performed on two CI engine vehicles with turbocharging and two SI engine vehicles that are naturally aspirated.

The first part of the article pays attention to the impact of clogged air filter on the course of the engine power and torque's curves.

The second part of the article shows the measurement results of the impact of clogged air filter on the fuel consumption and exhaust gas composition relating to the SI engine vehicle with air intake depending on atmospheric pressure. The measurements were also performed in extreme cases with air filter blanked off at 90% of its active surface as well as with forced air induction.

## 2. Materials and Methods

Measurements intended for the impact of air filter to be determined can be divided into the following points:

- measuring the course of the engine power and torque's curves
- measuring the vehicle's ability to accelerate
- measuring the fuel consumption and composition of the exhaust gases while driving at the constant speed
- conducting the emission control according to the current Slovak legislation

All the measurements were performed under laboratory conditions. The engine operating temperature was checked prior to measuring. The laboratory temperature varied from 19 °C up to 23 °C during measurements.

### 2.1. Vehicles and Air Filters Used during Measurements

Names and basic technical parameters of the vehicles used during measurements are given in Tables 1–4. There is also a level of fouling the filters given in Figures 1–4. While replacing the used air filters by the new ones, there was an attempt to use the filter of the same brand, specification and quality.

The measurements were performed by two CI engine vehicles with turbocharging and two SI engine vehicles with air intake drawn into the engine's cylinders depending on atmospheric pressure.

The technical parameters of the first vehicle used in measurements are given in Table 1.

**Table 1.** Technical data of Volkswagen (VW) Golf 7 [29].

| Name | VW Golf 7, 1.6 TDi, 77 kW |
|---|---|
| Engine | Compression ignition |
| Air intakes | Turbocharger, air-cooled |
| Fuel Injection Method | Common Rail |

Comparison of a new and clogged air filter relating to the first vehicle measured is shown in Figure 1.

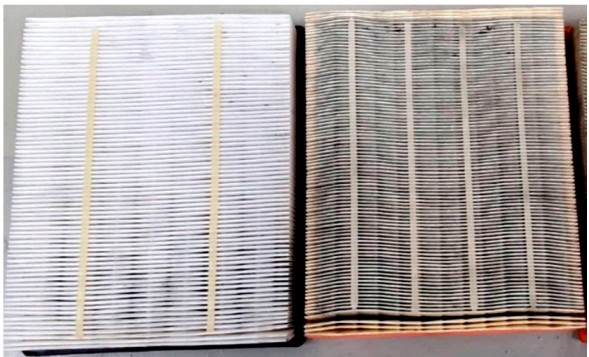

**Figure 1.** Air filters of VW Golf 7.

The second vehicle used had its data modified in the engine control unit in order to increase the amount of fuel injected as well as to increase the amount of air delivered into the cylinders via changes in turbocharger's vane geometry. The vehicle's technical data according to the manufacturer are given in Table 2.

**Table 2.** Technical data of VW Passat, 1.9 TDi [29].

| Name | VW Passat, 1.9 TDi, 74 kW—according to the manufacturer |
|---|---|
| Engine | Compression ignition |
| Air intakes | Turbocharger, air-cooled |
| Fuel Injection Method | Pump—jet |

The clogged and new air filters are shown in Figure 2.

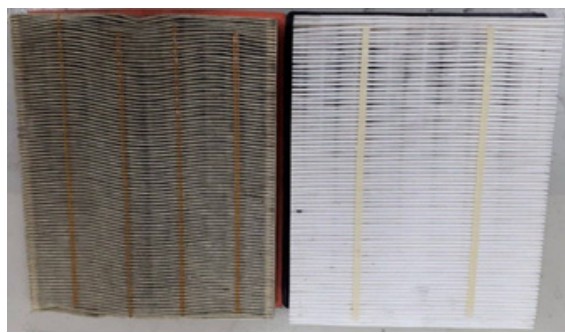

**Figure 2.** Air filters of VW Passat, 1.9 TDi.

The third vehicle measured was fitted with a spark-ignition engine with turbocharging and modified in the intake manifold and cylinder head, as well as having the data in the engine control unit modified in order to achieve better dynamic vehicle performance (See Table 3).

**Table 3.** Technical data of VW Golf 4 [29].

| Name | VW Golf IV, 1.8 5 V Turbo GTi–110 kW–according to the manufacturer |
|---|---|
| Engine | Spark-ignition |
| Air intakes | Turbocharger, air-cooled |
| Fuel Injection Method | Multi point injection |

Comparison of the new and clogged air filters are shown in Figure 3.

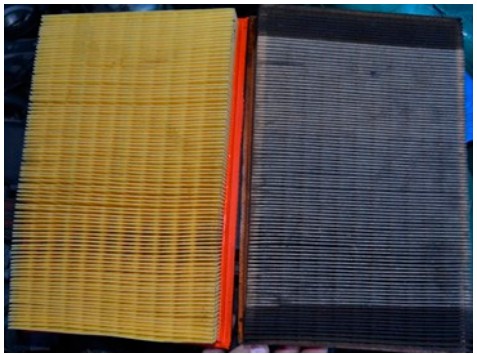

**Figure 3.** Air filters of VW Golf IV, 1.8.

The fourth vehicle used was fitted with a naturally aspirated spark-ignition engine, Table 4.

**Table 4.** Technical data of Kia Ceed, 1.6 CVVT [29].

| Name | Kia Ceed, 1.6 CVVT, 92 kW |
|---|---|
| Engine | Spark-ignition |
| Air intakes | Atmospheric |
| Fuel Injection Method | Multi point injection |

Figure 4 shows the new and clogged air filter used during measurement.

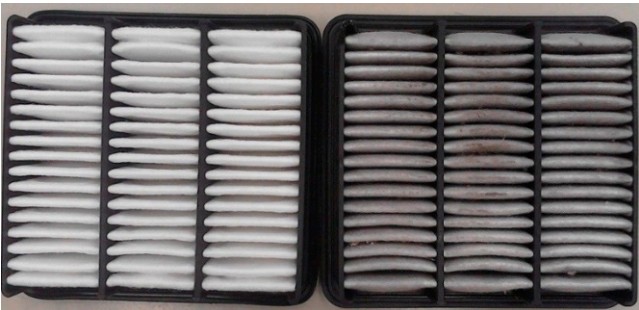

**Figure 4.** Air filters of Kia Ceed, 1.6 CVVT.

Concerning Kia Ceed, the measurements included situations in which the air filter's active surface had been blanked off at 90% and the air had been delivered into the intake duct through the ventilator. Figure 5 shows the attachment of the ventilator in front of the throttle.

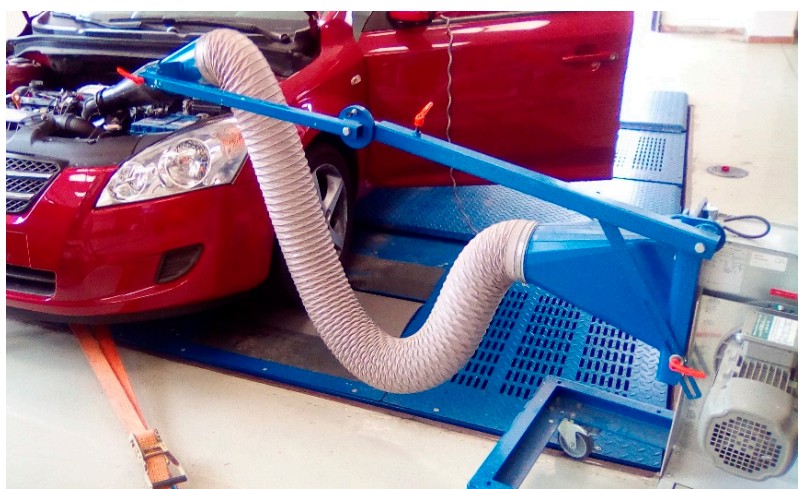

**Figure 5.** Attachment of ventilator in front of the throttle.

The maximum power of the ventilator was 2800 m$^3$ h$^{-1}$ [30]. The engine capacity of the vehicle used was 1.6 L. The maximum engine speed reached during the measurement was 6800 min$^{-1}$. After recalculating the engine speed frequency per hour and taking into account that only one from four piston upstrokes is the suction one, it can be said that the maximum theoretical mass, which can be delivered into the engine per hour with regard to capacity of cylinders, was 326 m$^3$. A higher amount of air delivered into the engine's cylinders would be dependent on their turbocharging. Thus, a ventilator can provide 8 times as much air delivered into the cylinders as the cylinders with atmospheric intakes are able to draw in for a period of an hour at the maximum engine speed.

### 2.2. Measuring the Impact of Clogged Air Filter on the Course of the Engine Power and Torque's

The purpose of measurements was to determine the impact of clogged air filter on the course of the engine power and torque's curves depending on the engine speed. These measurements were done on the cylinder power test station MAHA MSR 1050. The measurement deviation was ±2% from the overall value of power measured. MAHA MSR 1050's functions include a calculation of change in power on the basis of the change in the ambient temperature and atmospheric pressure. Thus, a change in the temperature or pressure in the laboratory does not affect the measurement results [31].

During measurements, the vehicle gradually reached the maximum engine speed at the penultimate gear with the acceleration pedal fully applied, so the course of power's curve got on the wheels was determined. After reaching the maximum engine speed, the driver applied the clutch pedal having the penultimate gear engaged. Following this and through the coasting, the value of difference between

the power delivered to the wheels onto the dynamometer's cylinders and the power on the engine's flywheel can be determined, respectively. This value of difference between the power delivered onto the dynamometer's cylinders and the power on the engine output is a power dissipation caused by mechanical losses [32].

### 2.3. Measuring the Vehicle Acceleration

Time needed for vehicle acceleration from 40 km h$^{-1}$ up to the speed of 120 km h$^{-1}$ was measured by MAHA MSR 1050 dynamometer. The measurements were performed at the fourth gear used with having the acceleration pedal fully applied as well as with forced air induction through the ventilator, with new and clogged air filters, and with air filter blanked off at 90% of its surface.

In order to carry out the measurements, it was necessary to enter the values of driving resistances from the vehicle used into the dynamometer's control computer. The values were obtained by the coasting deceleration measurement of vehicle resistance under the conditions of Standard EN 30 0556 [33]. Such measurement relies on a vehicle with prescribed laded mass which is accelerated up to the speed of about 120 km h$^{-1}$; disconnection between the engine and wheels, and on the recording of vehicle coasting. Figure 6 depicts the vehicle deceleration during measurement.

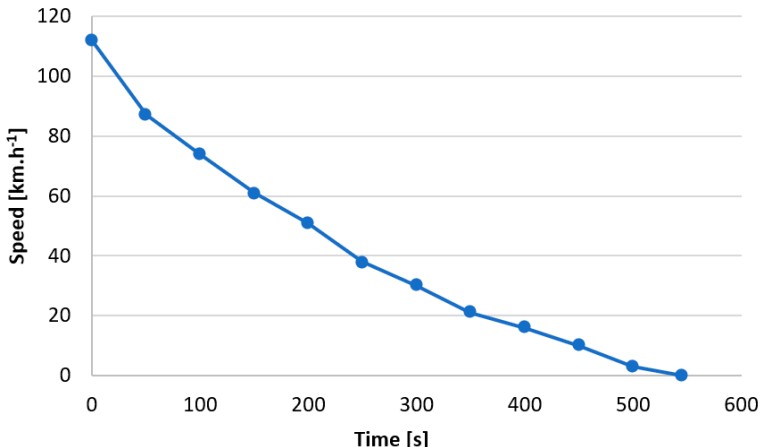

**Figure 6.** Driving speed during coasting deceleration measurement of vehicle resistance.

The recorded vehicle speed, which depends on the time of disconnection between the engine and wheels, was further introduced into the cylinder test station's computer. The coasting deceleration measurement of vehicle resistance was further repeated on the dynamometer, bearing in mind the change in the rolling resistance due to the dynamometer's cylinder rounding off. Thus, the cylinder test station can fully provide a road driving simulation.

The value of vehicle acceleration has been calculated from the difference of initial and final vehicle speed and time needed for the speed to be changed.

### 2.4. Measuring the Fuel Consumption and Composition of Exhaust Gases

The fuel consumption and composition of exhaust gases were measured with Kia Ceed fitted with forced air induction, new and clogged air filters as well as with the air filter being blanked off. At the same time, these were measured at a driving speed of 50 km h$^{-1}$ with the fourth gear engaged and at 90 km h$^{-1}$ and 130 km h$^{-1}$ with the fifth gear engaged. The measurements were applied when driving on the level ground and with 6% of a rising gradient.

The fuel consumption was determined by MAHA MSR 1050 dynamometer and by volume flow meter AIC 1203 with a measurement deviation of ±0.5% [34]. Figure 7 shows how the flow meter AIC 1203 was connected.

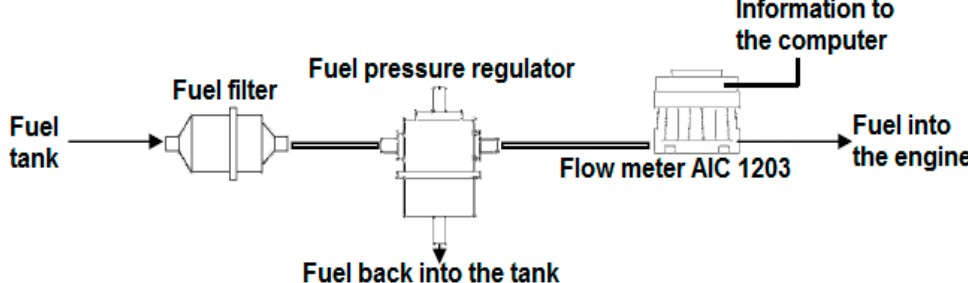

**Figure 7.** Connection of flow meter AIC 1203 into the vehicle fuel system during measurement.

Based on the fuel consumption given in [l.100 km$^{-1}$], the power delivered onto the dynamometer's cylinders and their transmission efficiency, it is possible to calculate the specific fuel consumption.

The amount of fuel consumed needed for 100 km to be driven can be determined by the cylinder test station's computer.

Based on the coasting deceleration measurement of vehicle resistance carried out when determining the course of the engine power and torque's curves, it is possible to calculate the transmission efficiency of power delivered onto the dynamometer's cylinders. If the transmission efficiency has a value of 75%, the power delivered onto the dynamometer's cylinders must be increased by 25% in order to calculate the value of power in the engine. The specific fuel consumption has been further calculated according to formula (1):

$$m_{pe} = (Q_{100} \cdot \rho)/(P_c \cdot t) \tag{1}$$

$m_{pe}$—brake-specific fuel consumption, BSFC, [g.kWh$^{-1}$]
$Q_{100}$—the amount of fuel needed for 100 km to be driven [l.100 km$^{-1}$]
$\rho$—fuel density [kg.m$^{-3}$]
$P_c$—power at the engine's output [kW]
t—time needed for 100 km to be driven [h] [35]

*2.5. Measuring the Impact of Air Filter on Vehicle's Ability to Meet the Emission Limits Resulting from Legislation*

The composition of exhaust gases was measured by the exhaust gas analyzer MAHA MGT 5 that analyzes, in particular, the lambda coefficient and volume of CO, $CO_2$ and HC.

The emission analyzer MAHA MGT 5 assessed the vehicle's fitness for road driving while having the forced air induction, new and clogged air filters and when using the air filter blanked off at 90% of its surface. The exhaust gas composition was determined through the vehicle coasting and increased speed according to [36]. Measurements were performed with Kia Ceed, see Table 4.

The values that must be complied are given in Table 5.

**Table 5.** Legislative requirements on concentration of exhaust gas components [36].

| Parameter | Engine Speed [min$^{-1}$] | CO | HC | λ | Engine Diagnostics Light |
|---|---|---|---|---|---|
| **Coasting** | 510–810 | 0.40 | 100 | unassessed | Off |
| **Increased Speed** | 2500–3000 | 0.30 | 100 | 0.97–1.03 | Off |

Besides the parameters from Table 5, the values of $CO_2$ and $O_2$ were assessed as well.

## 3. Results

*3.1. Results from Measuring the Impact of Clogged Air Filter on the Course of the Engine Power and Torque's Curves*

The course of the engine power and torque's curves relating to the first vehicle tested (Volkswagen (VW) Golf 7) is shown in Figure 8. The red curve depicts the course of engine power and the blue curve depicts the course of power on the wheels. The green curve depicts the course of power dissipation. The orange curve represents the course of engine torque. Thick curves represent the measurements with clogged air filter, and thin curves with the new one.

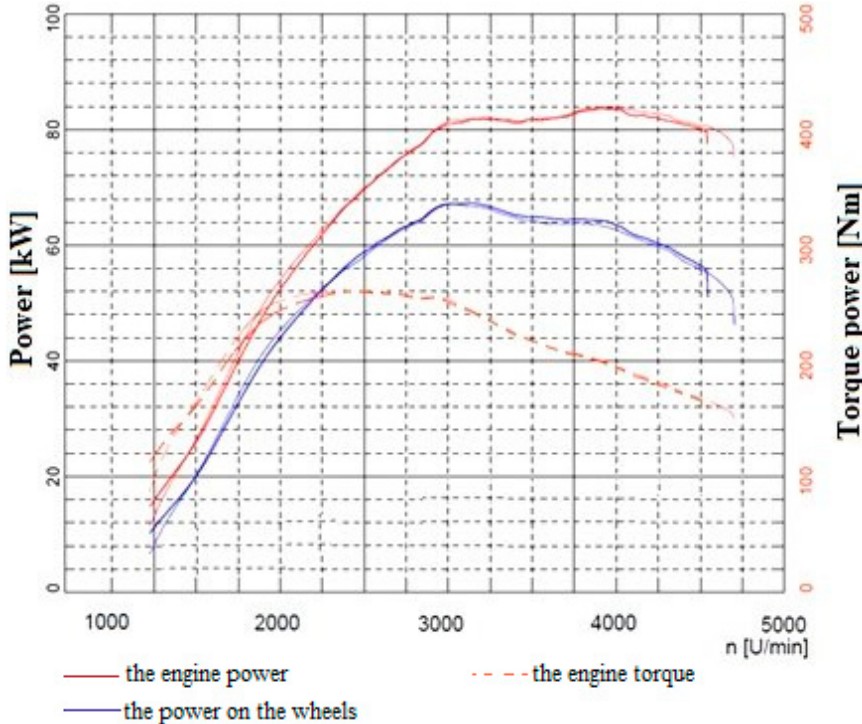

**Figure 8.** Measurement results with VW Golf 7.

As seen from Figure 8, the course of curves is almost identical. The differences in the beginning are irrelevant since they could be inaccuracies caused by a time needed for the acceleration pedal to be fully applied. The greatest differences are seen in the first third of the engine speed.

The intake manifold of the VW Golf 7 was also fitted with an air mass meter, MAF, and thus it was able to measure and record the intake air mass while measuring the engine power and torque. Figure 9 shows the course of intake air mass depending on the engine speed. The intake air mass is given in $g \cdot s^{-1}$.

When comparing the curves from Figures 8 and 9, it is possible to see a high correlation value between the curves of engine power and torque and the curves representing the intake air mass. However, both cases show slight differences.

The measurement results with the VW Passat, 1.9 Tdi are shown in Figure 10. Thin curves represent measurements with new air filter and thick curves with the clogged one.

The differences measured with this second vehicle were relatively small. The greatest differences were measured in the first third of the engine speed as was the case with the VW Golf 7.

The third vehicle measured was the VW Golf 4. Figure 11 depicts the course of the engine power and torque's curves depending on the air filter. Thick curves represent measurements with new air filter and thin curves with the clogged one.

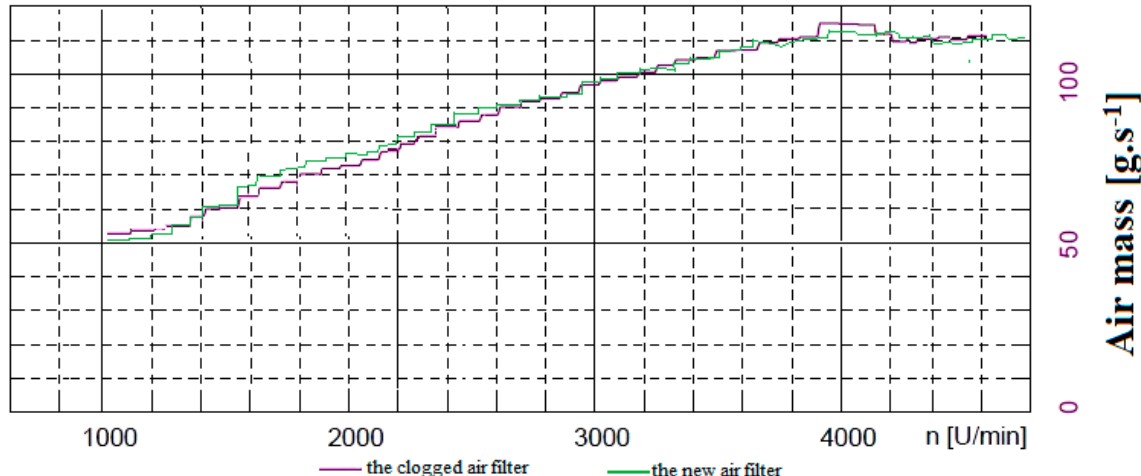

**Figure 9.** The intake air mass during measurement with VW Golf 7.

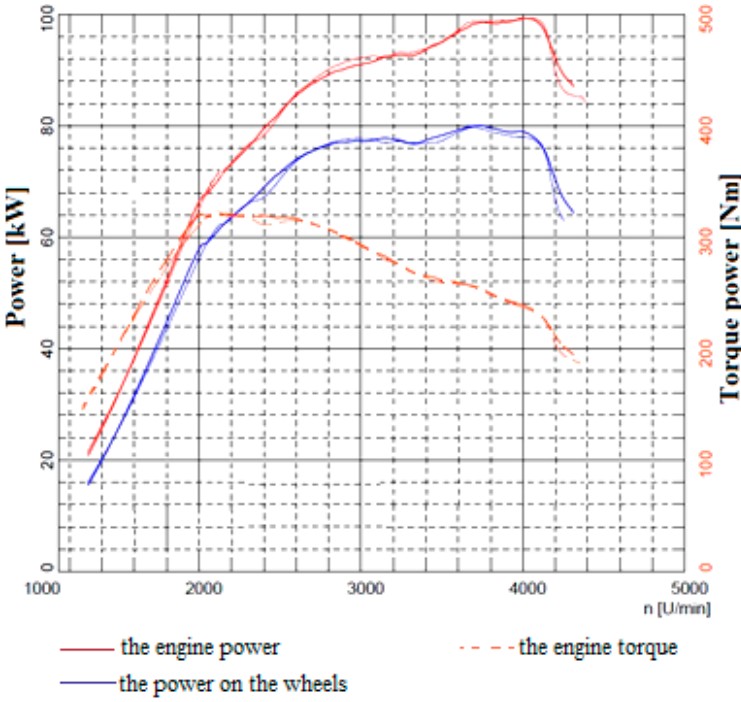

**Figure 10.** Measurement results with VW Passat.

The differences within the VW Golf 4 with spark-ignition engine that is naturally aspirated were measured predominantly in the upper half of the engine speed.

Figure 12 depicts the course of the engine power and torque's curves measured with the KIA Ceed. Thin line indicates the curves measured within the forced air induction, long dashed line belongs to measurements with new air filter, thick line represents the curves with clogged air filter and short dashed line belongs to the curves with air filter blanked off at 90% of its active surface.

As seen from Figure 12, the strongest differences are in the upper half of the engine speed, like in Figure 11. The greatest differences were measured during measurements with the air filter that was blanked off and with forced air induction.

For better transparency, the results from Figure 12 are also given in Table 6.

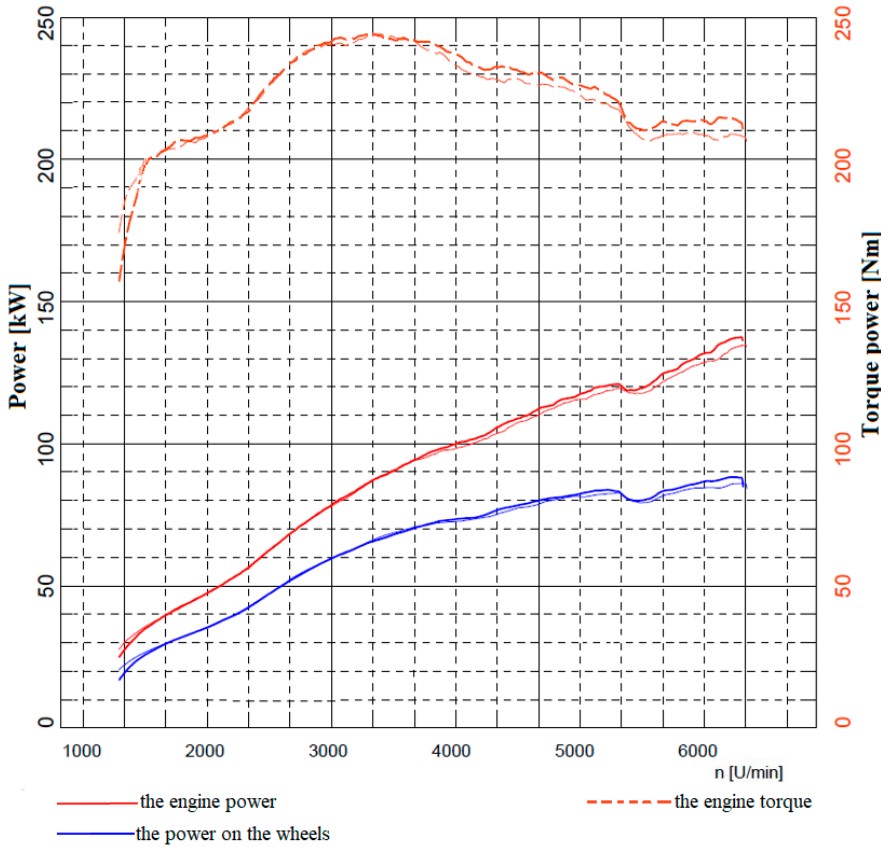

**Figure 11.** Measurement results with VW Golf 4.

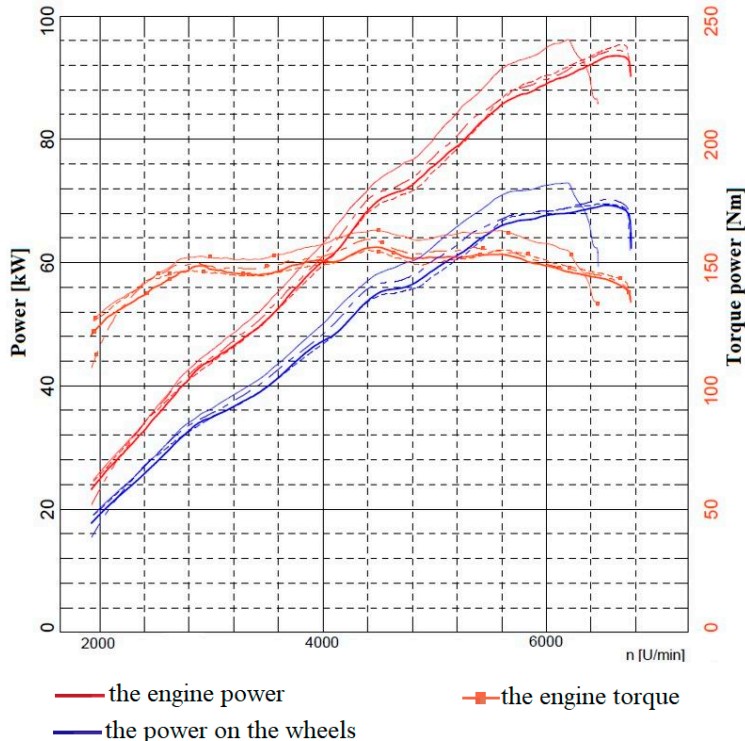

**Figure 12.** Measurement results with Kia Ceed.

**Table 6.** Impact of air filter on the engine power and torque power.

| Vehicle | Air filter | Maximum Engine Power [kW]/[min⁻¹] | Maximum Engine Torque [Nm]/[min⁻¹] | The Greatest Difference in the Engine Power [kW]/[min⁻¹] | The Greatest Difference in the Engine Torque [Nm]/[min⁻¹] |
|---|---|---|---|---|---|
| VW Passat | New | 98/4010 | 321/2015 | | |
| VW Passat | Used | 98/4010 | 324/2050 | 2/2860 | 9/2405 |
| VW Golf 7 | New | 84/4200 | 260/2400 | | |
| VW Golf 7 | Used | 83/4200 | 258/2400 | 5/1750 | 13/2000 |
| VW Golf 4 | New | 138/6326 | 233/3334 | | |
| VW Golf 4 | Used | 136/6326 | 234/3334 | 4/6323 | 6/6320 |
| Kia Ceed | Forced air induction | 97/6210 | 164/4405 | | |
| Kia Ceed | New | 95/6562 | 159/4415 | | |
| Kia Ceed | Used | 94/6659 | 156/4430 | 6/6210 | 12/5600 |
| Kia Ceed | Blanked off | 93/6645 | 154/4405 | | |

As seen from Table 6, the overall differences were of relatively low values. Concerning the CI engines with turbocharging, the greatest differences were in the first third of the engine speed; concerning the SI engines that were naturally aspirated, such differences were found in the upper half of the engine speed.

### 3.2. Results from Measuring the Vehicle Acceleration

Table 7 shows the time needed for the Kia Ceed's acceleration from a speed of 40 km.h⁻¹ up to 120 km.h⁻¹ as well as this acceleration depending on the air filter.

**Table 7.** Impact of air filter on vehicle acceleration.

| Air Filter | Time Needed for Acceleration [s] | Acceleration [m.s⁻²] |
|---|---|---|
| **Forced Air Induction** | 18.531 | 1.199 |
| **New** | 18.692 | 1.189 |
| **Used** | 19.574 | 1.135 |
| **Clogged** | 19.684 | 1.129 |

Time needed for vehicle acceleration and value of vehicle acceleration are similar for both forced air induction and new air filter. While measuring with the air filter used and blanked off, the vehicle's ability to accelerate decreases. The biggest difference, 1.153 s, was measured between the measurements with forced air induction and blanked-off air filter. This kind of difference, albeit small, can also be seen while driving.

### 3.3. Results from Measuring the Fuel Consumption and Composition of Exhaust Gases

Table 8 contains the results from measuring the fuel consumption while driving at the constant speeds of 50 km.h⁻¹, 90 km.h⁻¹ and 130 km.h⁻¹. The measurement was performed with the Kia Ceed.

**Table 8.** Impact of air filter on fuel consumption.

| Vehicle Speed | 50 km.h⁻¹ | | 90 km.h⁻¹ | | 130 km.h⁻¹ | |
|---|---|---|---|---|---|---|
| **Road Rising Gradient** | 0% | 6% | 0% | 6% | 0% | 6% |
| **Forced Air Induction** | 3.2 | 8.0 | 4.2 | 11.6 | 7.0 | 15.8 |
| **New** | 3.0 | 8.0 | 4.2 | 11.5 | 6.9 | 15.7 |
| **Used** | 3.1 | 7.9 | 4.2 | 11.5 | 6.9 | 15.8 |
| **Blanked Off** | 3.2 | 7.9 | 4.2 | 11.6 | 7.0 | 15.8 |

As seen from Table 8, the difference in fuel consumption is small, almost reaching the level of measurement deviation. However, it can be seen that the highest fuel consumption in all the results was

measured within the forced air induction and blanked-off filter. However, there are some differences in low values in fuel consumption.

The value of power delivered into the dynamometer's cylinders was measured as well. The results are given in Table 9. Since this kind of power is not affected by the air filter, the results apply for all the measurements together.

**Table 9.** The power delivered into the dynamometer's cylinders.

| Vehicle Speed | 50 km.h$^{-1}$ | | 90 km.h$^{-1}$ | | 130 km.h$^{-1}$ | |
|---|---|---|---|---|---|---|
| Road Rising Gradient | 0% | 6% | 0% | 6% | 0% | 6% |
| The Power Delivered into the Dynamometer's Ccylinders [kW] | 2 | 11 | 4.3 | 25 | 16 | 45 |

Based on data from Tables 8 and 9, the specific fuel consumption has been calculated and it is given in Table 10 in [g.kWh$^{-1}$].

**Table 10.** Impact of air filter on specific fuel consumption.

| Vehicle Speed | 50 km.h$^{-1}$ | | 90 km.h$^{-1}$ | | 130 km.h$^{-1}$ | |
|---|---|---|---|---|---|---|
| Road Rising Gradient | 0% | 6% | 0% | 6% | 0% | 6% |
| Forced Air Induction | 480 | 218 | 537 | 251 | 341 | 274 |
| New | 450 | 218 | 527 | 248 | 336 | 272 |
| Used | 465 | 215 | 527 | 248 | 336 | 274 |
| Blanked Off | 480 | 215 | 527 | 251 | 341 | 274 |

The dependency of specific fuel consumption on air intakes seen in Table 10 is the same as in Table 8 since there is, in this case, a mathematical dependency between the amount of fuel consumed per certain distance and the value of specific fuel consumption [34].

The volume of CO in the exhaust gases during fuel consumption measurement is given in Table 1 in [%].

The air filter has had a relatively significant effect on the CO content in the exhaust gases as shown in Table 11.

**Table 11.** Impact of air filter on volume of CO in the exhaust gases.

| Vehicle Speed | 50 km.h$^{-1}$ | | 90 km.h$^{-1}$ | | 130 km.h$^{-1}$ | |
|---|---|---|---|---|---|---|
| Road Rising Gradient | 0% | 6% | 0% | 6% | 0% | 6% |
| Forced Air Induction | 0.11 | 0.37 | 0.25 | 0.42 | 0.35 | 0.81 |
| New | 0.13 | 0.37 | 0.27 | 0.42 | 0.36 | 0.82 |
| Used | 0.43 | 0.42 | 0.35 | 0.51 | 0.38 | 0.90 |
| Blanked Off | 0.43 | 0.42 | 0.41 | 0.52 | 0.52 | 0.92 |

Table 12 shows the results from measuring the $CO_2$ concentration in the exhaust gases. This $CO_2$ concentration is given [%].

**Table 12.** Impact of air filter on volume of $CO_2$ in the exhaust gases.

| Vehicle Speed | 50 km.h$^{-1}$ | | 90 km.h$^{-1}$ | | 130 km.h$^{-1}$ | |
|---|---|---|---|---|---|---|
| Road Rising Gradient | 0% | 6% | 0% | 6% | 0% | 6% |
| Forced Air Induction | 14.9 | 14.9 | 14.8 | 15 | 15 | 14.8 |
| New | 14.9 | 14.8 | 14.9 | 15 | 14.9 | 14.7 |
| Used | 14.9 | 14.9 | 14.8 | 14.9 | 15 | 14.8 |
| Blanked Off | 14.9 | 14.9 | 15 | 14.5 | 14.9 | 14.9 |

Relating to the air filter, driving speed and road rising gradient, the value of $CO_2$ concentration in the exhaust gases was almost invariable.

The HC concentration in the exhaust gases is shown in Table 13 and its value is given in [ppm].

**Table 13.** Impact of air filter on volume of HC in the exhaust gases.

| Vehicle Speed | 50 km.h$^{-1}$ | | 90 km.h$^{-1}$ | | 130 km.h$^{-1}$ | |
|---|---|---|---|---|---|---|
| Road Rising Gradient | 0% | 6% | 0% | 6% | 0% | 6% |
| Forced Air Induction | 6 | 16 | 7 | 4 | 9 | 3 |
| New | 7 | 15 | 6 | 4 | 8 | 3 |
| Used | 6 | 16 | 6 | 5 | 6 | 2 |
| Blanked Off | 7 | 12 | 3 | 4 | 5 | 2 |

The value of air-fuel ratio λ during measurements is shown in Table 14.

**Table 14.** Impact of air filter on volume of λ.

| Vehicle Speed | 50 km.h$^{-1}$ | | 90 km.h$^{-1}$ | | 130 km.h$^{-1}$ | |
|---|---|---|---|---|---|---|
| Road Rising Gradient | 0% | 6% | 0% | 6% | 0% | 6% |
| Forced Air Induction | 1.001 | 1.002 | 1.001 | 1.001 | 1.001 | 0.870 |
| New | 1.000 | 1.001 | 1.000 | 1.001 | 1.001 | 0.880 |
| Used | 1.000 | 1.000 | 1.001 | 0.999 | 1.002 | 0.890 |
| Blanked Off | 1.000 | 1.001 | 1.000 | 1.000 | 1.003 | 0.890 |

The fuel mixture composition has been maintained stoichiometric throughout the entire period via control unit, outside the conditions of high motor load, during a road rising gradient of 6% with driving speed of 130 km.h$^{-1}$ when the mixture has been rich (Table 14).

*3.4. Results from Measuring the Impact of Air Filter on Vehicle's Ability to Meet the Emission Limits Resulting from Legislation*

Table 15 shows the values measured during emission control of the Kia Ceed. For better transparency, the measured values that exceed the authorised limit are given in red.

**Table 15.** The values measured during emission control of Kia Ceed.

| Parameter | CO | | CO$_2$ | | HC | | O$_2$ | | λ | |
|---|---|---|---|---|---|---|---|---|---|---|
| Forced Air Induction | 0.02 | 0.01 | 14.91 | 14.83 | 45 | 41 | 0.29 | 0.14 | 1.011 | 1.004 |
| New | 0.01 | 0.00 | 15.13 | 14.84 | 46 | 30 | 0.14 | 0.07 | 1.004 | 1.002 |
| Used | 0.01 | 0.01 | 15.19 | 14.86 | 16 | 32 | 0.13 | 0.07 | 1.004 | 1.001 |
| Blanked Off | 0.11 | 0.46 * | 14.97 | 14.12 | 136 * | 119 * | 0.07 | 0.68 | 0.994 | 1.014 |

\* For better transparency, the measured values that exceed the authorised limit are noted.

As seen from the comparison of Tables 5 and 15, the vehicle with air filter blanked off was assessed as unfit for road driving. The vehicle engine produced higher than permitted values of CO and HC.

## 4. Discussion

Concerning the compression ignition engine with turbocharging, the effect of air intakes on the course of engine power and torque's curves is relatively low as seen in Figures 8 and 10, and in publication [37] which focuses its attention to the impact of air filter on selected operating parameters of a vehicle with CI engine with turbocharging as well. By using two different filters, there was a stronger difference obtained within the stationary engine seen in publication [38]. However, the engine was naturally aspirated, not turbocharged. Concerning the CI engine without turbocharging, there was a greater difference measured also in the case seen in publication [39] from which the results are shown

in Figure 13. The measurements were being performed with a new, clogged air filter and without the air filter. The specific fuel consumption was measured as well.

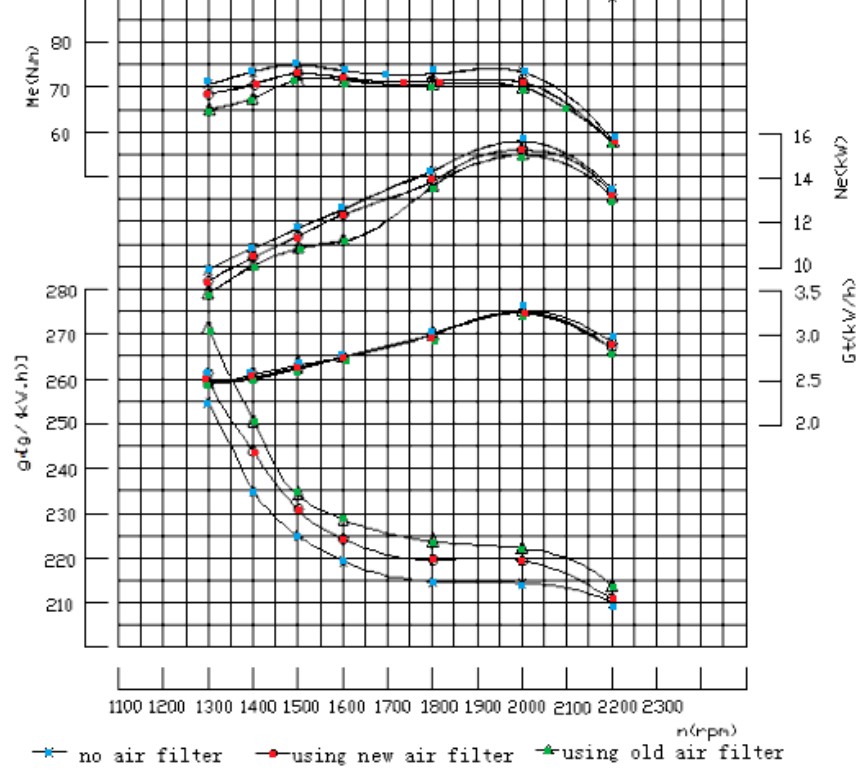

**Figure 13.** Impact of air filter within naturally aspirated compression ignition (CI) engine [39].

As seen from Figure 13, the air filter affected the course of the engine power and torque's curves as well the fuel consumption. Concerning the differences measured, the percentage is higher than in Figures 8 and 9. One of the reasons is probably the way the air is delivered into the cylinders—by turbocharger (Figures 8 and 9) or by being naturally aspirated (Figure 13). In the case from publication [39], change in the air filter, respectively its absence, led to an increase in temperature of the exhaust gases by approximately 20 °C, as seen in Figure 14 as well.

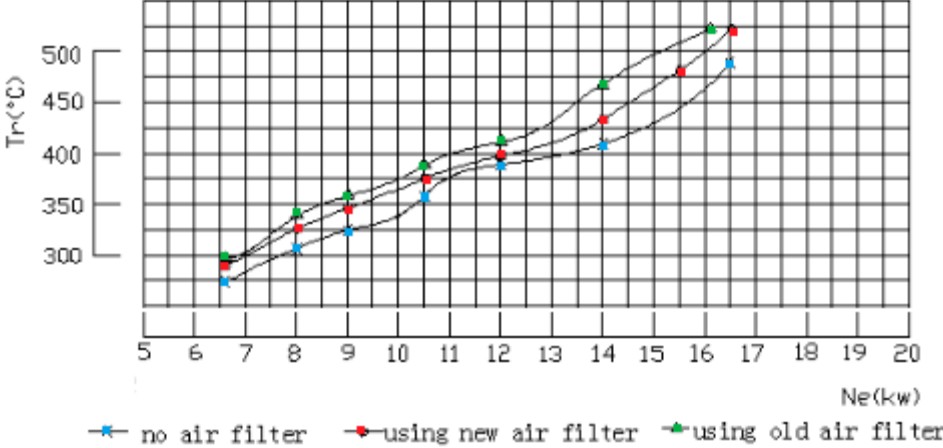

**Figure 14.** Impact of air filter on change in the exhaust gases' temperature [39].

Such a change in the exhaust gases' temperature (Figure 14) was presumably due to change in a mixture's richness as well as in the length of its burning. The temperature of the exhaust gases was not

being measured during measurements needed for this publication. However, even though this kind of measurement would have been performed, it would probably not lead to a change in the exhaust gases' temperature. The measurements were performed with the engine with a turbocharger that largely offset the change in mechanical resistance of the air filter. When measuring with the SI engine as well, it was found that the value of λ had been changed only under extreme conditions, as also seen in Table 14. Apart from the extremes, the exhaust gases' temperature would probably remain the same.

The effect of forced air induction into the engine on the course of engine power and torque's curves within the engine that is naturally aspirated has been also explored in publication [40]. The measurements were performed with the engine with comparable capacity, with a ventilator of low performance placed into the vehicle's intake manifold. The results of impact of forced air induction on the course of the engine power and torque's curves are depicted in Figure 13. The solid line represents the results measured while using the ventilator and the dashed line indicates the results without using it.

Placing the ventilator of low performance into the intake manifold increased the value of engine torque in relatively short extent of the engine speed (Figure 15), whereas the forced air induction through the ventilator of high performance evinced in the engine speed fully (Figure 12). Within the higher engine speed, the value of engine torque decreased due to forced induction. This is probably a result of deficient ventilator performance since the ventilator placed in the engine inlet manifold did not deliver the air, but reduced the air flow [40]. A ventilator of sufficient performance has a potential to increase the engine's volumetric efficiency, and can lead to an overall increase in the engine power and torque, as also seen in publication [41]. Using the ventilator with power of about 750 W led to an increase in volumetric efficiency by up to 9%. A condition for the engine's volumetric efficiency and for the engine power and torque to be increased, within the engines controlled by control unit, is reprogramming of the engine's control unit, as also shown in publications [42–44].

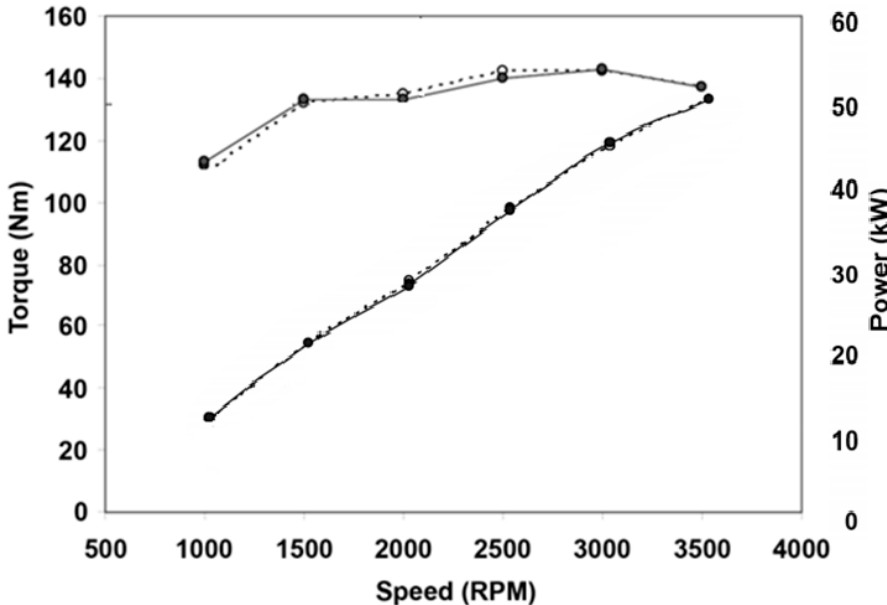

**Figure 15.** The effect of forced air induction on the engine power and torque [40].

Most likely, the greater impact of air filter on the course of the engine power and torque's curves would be while measuring with a carburettor vehicle as seen in publications [44,45].

The results from measuring the fuel consumption and vehicle emission depending on the air filter are broadly equivalent to the results shown in publication [46]. These results also point to a relatively low effect of air filter on fuel consumption and emission composition within the naturally aspirated SI engine vehicles. There were also minor changes measured within the vehicle's ability to accelerate. The measurements were conducted with the SI engine vehicle with turbocharging as well, however,

the differences were small [46]. Greater differences were measured only with a carburetted vehicle from 1972. The substantial increase in the engine power and torque, almost by 50% in some spectrum of the engine speed, was also found within the SI engine with a carburettor that was additionally fitted with a turbocharger, as described in publication [47]. For a rough comparison, when relating to the engine with additionally fitted turbocharger, the measurements, during which the ventilator had been placed in front of the intake manifold, can be used, as shown in Figure 5. Here, it led to a significant lower increase in the engine power and torque, as also seen in publication [47]. The reason lies also in the fact that there was the engine with a control unit used during measurements relating to this article, whereas, in relation to publication [47], there was the engine with carburettor used. Therefore, the higher air mass caused the increase in the amount of fuel taken by the air in the engine's carburettor.

## 5. Conclusions

The research paid attention to the impact of air intakes on selected parameters of different vehicles.

The first parameter covered the impact of air filter on the course of the engine power and torque's curves. Concerning the CI engine with turbocharging, only a low impact of clogged air filter on the course of the engine power and torque's curves was seen. Within both vehicles, such an impact has shown in the engine speed of about the same value. It is probably a result of turbocharger's control management since the differences were measured predominantly in the area of engine speed that is outside the turbocharger's stronger effect [48]. The differences of 9 Nm relating to the first vehicle and 13 Nm relating to the second vehicle are relatively small, also due to the fact that the differences were measured in the short extent of engine speed. Such a difference is negligible in common road traffic [49].

The impact of clogged air filter within the SI engine vehicles was low as well. Concerning the VW Golf IV, the difference was 6 Nm and only 4 Nm for the Kia Ceed. Both differences were measured in almost identical area of engine speed.

The greater difference was achieved when comparing the forced air induction and air filter blanked off at 90% of its surface. In this case, the difference was 12 Nm in substantially greater extent of the engine speed. Together with the value of engine torque's difference, the extent of engine speed at which the change has been monitored is also important.

The differences in the courses of curves from Figure 12 have also occurred during vehicle acceleration. Such differences can be noticed in a common road driving, even though higher differences may be caused, for instance by the distance and speed of wind, the road rising/falling gradient and so on [50].

The increased fuel consumption has been measured in two cases. The first one was forced air induction and the second one was a blanked-off air filter. This probably happened due to reduction of engine energy efficiency [51]. Concerning the forced induction, the most likely reason lies in lowering the angle of throttle opening due to air overpressure entering the throttle [52]. Despite the mixture being stoichiometric as shown by $\lambda$ value (Table 14), the engine control unit was not programmed for air overpressure. Thus, it can result in false value of injected mass of fuel, ignition timing, timing of valves or other parameters affecting the energy efficiency. However, the increase in fuel consumption has been at a low level [53]. The change in fuel consumption by 0.1 L 100 km$^{-1}$ can be caused by several factors while driving, and the change in air intakes would probably have no effect on fuel consumption while real road driving [54].

While measuring the fuel consumption, the volume composition of exhaust gases was also determined. The strongest difference was measured at the CO concentration since there was an increase due to blanking off the air filter. The air intake affected the CO concentration mostly at low motor load. By increasing motor load due to rising gradient and driving speed, such an impact was reduced.

Research has shown the effect of air intakes on a vehicle's ability to meet the legislative emission requirements as well. When using the air filter blanked off, a vehicle was inadequate for these legislative requirements and declared unfit for road driving. The CO concentration was exceeded at higher engine

speed, and the HC concentration even at both coasting and higher speed. However, λ value was at its prescribed interval, implying continued combustion of stoichiometric mixture.

Although there has been a relatively low impact of air filter on selected vehicle parameters, replacing the air filter should not be neglected. Since the air filter does not serve for increasing the engine power, it captures impurities that could significantly reduce the engine lifespan.

**Author Contributions:** Conceptualization: F.S., J.S., methodology: F.S., J.S., formal analysis: A.K., data curation: F.S., writing—original draft preparation: F.S., writing—review and editing: A.K., project administration: A.K., funding acquisition: A.K. All authors have read and agreed to the published version of the manuscript.

**Funding:** This research was funded by the Slovak scientific grant agency VEGA of the Ministry of Education—no. 1/0436/18—Externalities in road transport, an origin, causes and economic impacts of transport measures.

**Conflicts of Interest:** The authors declare no conflict of interest.

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
