# Peer review of "Air Filter and Selected Vehicle Characteristics"

_sustainability, doi:10.3390/su12229326_

Round 1

Reviewer 1 Report

DEar authors,

The paper entitled “Air Filter and Selected Vehicle Characteristics” focuses on the impact of clogged air filter on selected vehicle characteristics such as dynamic features, fuel consumption and composition of exhaust gases. The topic of this paper is of interest for the Journal “Sustainability” and can be pubblished after some revisions as follows:

  • Please pay attention that all units are expressed in the IS;
  • Line 184: fuel consumption uniti is wrong;
  • Please number all equations and report the numbers in text when cited;
  • Figures 8,9,10 is poor quality, it would be better to improve the qualitiy;
  • Table 5 preents some numbers in red.

Author Response

Dear reviewer, 

thank you for your review.

The paper entitled “Air Filter and Selected Vehicle Characteristics” focuses on the impact of clogged air filter on selected vehicle characteristics such as dynamic features, fuel consumption and composition of exhaust gases. The topic of this paper is of interest for the Journal “Sustainability” and can be pubblished after some revisions as follows:

  • Please pay attention that all units are expressed in the IS; - Revised as the reviewer suggested
  • Line 184: fuel consumption uniti is wrong; The fuel consumption unit in paper is:

[l.100 km-1]. Unit [l.100 km-1] is the usual unit for expressing fuel consumption over a given distance. According to SI, the expression would be in [m-3.100 000m-1], but such an expression is not usual at all. Does the reviewer have a proposal for another unit that expresses the amount of fuel consumed over a distance?

  • Please number all equations and report the numbers in text when cited; Revised as the reviewer suggested
  • Figures 8,9,10 is poor quality, it would be better to improve the qualitiy; Revised as the reviewer suggested
  • Table 5 preents some numbers in red. Probably Table 15. The numbers are red intentionally - explained in "346". For better transparency, the measured values that exceed the authorised limit are given in red.

 Sincerely

             Author.

Reviewer 2 Report

46 Please explain briefly in the text what the abbreviations SI and CI mean

In my opinion, very important information is missing that could influence the measurement results. What brand were the new and dirty filters? Is their quality comparable?

Is there one curve or two in Figure 9?

276 The times are similar, but the maximum difference is as much as 6%. In my opinion, this is a big difference that can be noticeable during normal driving.

What is the error in measuring fuel consumption? The fuel consumption differences shown in Table 8 are very small. It is difficult to conclude here that the car consumes the most fuel in two cases.

Nowhere can I see information about the outside temperature during all measurements. Was each measurement performed under the same conditions?

Were the car engines hot during each of the measurements?

Interesting work, but without the above information, the results may be invalid.

Author Response

Dear reviewer, 

thank you for your review.

46 Please explain briefly in the text what the abbreviations SI and CI mean - Revised as the reviewer suggested, 50

In my opinion, very important information is missing that could influence the measurement results. - Revised as the reviewer suggested, 153 – 167, 93 – 94, 196, 97 - 99

What brand were the new and dirty filters? Is their quality comparable? - Revised as the reviewer suggested, 97 - 99

Is there one curve or two in Figure 9? Two, 249 - 250

276 The times are similar, but the maximum difference is as much as 6%. In my opinion, this is a big difference that can be noticeable during normal driving. Revised as the reviewer suggested, 294 – 298,

What is the error in measuring fuel consumption? ± 0.5%, 196

The fuel consumption differences shown in Table 8 are very small. It is difficult to conclude here that the car consumes the most fuel in two cases. Revised as the reviewer suggested, 304 - 307

Nowhere can I see information about the outside temperature during all measurements. Revised as the reviewer suggested, 93 - 94

 Was each measurement performed under the same conditions? Yes, 154 – 159,

Were the car engines hot during each of the measurements? Yes, 93 - 93

Interesting work, but without the above information, the results may be invalid. Revised as the reviewer suggested , 154 – 159, 93 – 94, 196, 97 - 99

Sincerely

               Author.

Reviewer 3 Report

In the manuscript ID: sustainability-947294, entitled “Air filter and selected vehicle characteristics”, the authors determined the impact of a clogged air filter on selected vehicle characteristics. In general, the manuscript is of interest, even if the design of the study is not immediately understandable. Detailed suggestions are reported below.

  • Abstract: I suggest to the authors including some more specific results in this part.
  • Line 46: Can authors better specify the “CI” and “SI” engine.
  • Materials and methods: I don't know if it is possible but if it, I suggest using the same sub-section title reported in the materials and methods and in the results (some look very similar).
  • Materials and methods: I suggest editing the materials and methods section. In my opinion, the experimental design of the study is not immediately understandable.
  • Tables 1-4 and figures 1-4: If possible, I would suggest merging the figures and tables as much as possible.
  • Figure 8, 10, 11, 12: I suggest to add a legend in the figure (as well as in other figures) and to replace the orange color (or the red one) with another one (the red line merges with the orange one).
  • Paragraph 3.3.: I suggest to merge the tables (8-15).
  • Discussions: In my opinion, this paragraph is not exhaustive and I suggest to expand.

Author Response

Dear reviewer, 

thank you for your review.

 In the manuscript ID: sustainability-947294, entitled “Air filter and selected vehicle characteristics”, the authors determined the impact of a clogged air filter on selected vehicle characteristics. In general, the manuscript is of interest, even if the design of the study is not immediately understandable. Detailed suggestions are reported below.

  • Abstract: I suggest to the authors including some more specific results in this part. Revised as the reviewer suggested, 17 - 20
  • Line 46: Can authors better specify the “CI” and “SI” engine. Revised as the reviewer suggested, 50
  • Materials and methods: I don't know if it is possible but if it, I suggest using the same subsection title reported in the materials and methods and in the results (some look very similar). Revised as the reviewer suggested, 153, 168, 224, 234, 288, 343
  • Materials and methods: I suggest editing the materials and methods section. In my opinion, the experimental design of the study is not immediately understandable. Revised as the reviewer suggested, 85 – 91, 95 - 153
  • Tables 1-4 and figures 1-4: If possible, I would suggest merging the figures and tables as much as possible. In my opinion, it is not appropriate to merge these Figures and Tables. Particular Tables have a different number of lines, columns and they express various quantities. If they merge, it will be very opaque from my point of view.
  • Figure 8, 10, 11, 12: I suggest to add a legend in the figure (as well as in other figures) and to replace the orange color (or the red one) with another one (the red line merges with the orange one). - Revised as the reviewer suggested, Figure 8, 9, 10, 11, 12
  • Paragraph 3.3.: I suggest to merge the tables (8-15). In my opinion, it is not appropriate to merge these Figures and Tables. Particular Tables have a different number of lines, columns and they express various quantities. One of them represents the matter relating to the engine's power, another one relates to the exhaust gases. If they merge, it will be very opaque from my point of view.
  • Discussions: In my opinion, this paragraph is not exhaustive and I suggest to expand. Revised as the reviewer suggested, 397 – 419, 435 – 441, 450 – 459.

 Sincerely

                Author.

Round 2

Reviewer 3 Report

The authors arranged the manuscript according to the suggestions and comments: I have no other comments regarding the manuscript.

Best regards